# Effect of bosentan in pulmonary hypertension development in systemic sclerosis patients with digital ulcers

Ivan Castellví[1☺]*, Carmen Pilar Simeón[2☺], Monica Sarmiento[1‡], Jordi Casademont[3‡], Hèctor Corominas[1‡], Vicenç Fonollosa[2‡]

1 Department of Rheumatology, Hospital Universitari de la Santa Creu i Sant Pau, Barcelona, Spain, 2 Department of Internal Medicine, Hospital Universitari de Vall Hebron, Barcelona, Spain, 3 Department of Internal Medicine, Hospital Universitari de la Santa Creu i Sant Pau, Barcelona, Spain

☺ These authors contributed equally to this work.
‡ These authors also contributed equally to this work.
* icastellvi@santpau.cat

**Data Availability Statement:** All relevant data are within the manuscript and supporting information file.

## Abstract

Systemic sclerosis is a disease where microcirculation damage is critical in their beginning and vascular complications have similar pathogenic findings. Digital ulcers are a frequent complication in systemic sclerosis patients and pulmonary hypertension is one of the leading causes of death. The use of bosentan has been shown to be useful for the treatment of pulmonary arterial hypertension and to prevent new digital ulcers. However, is unknown if bosentan can prevent pulmonary hypertension. Our objective was to determine if bosentan is useful to prevent pulmonary hypertension in SSc patients. A retrospective study in 237 systemic sclerosis patients with digital ulcers history treated or not with bosentan to prevent it was made. We analyzed the occurrence of pulmonary hypertension defined by an echocardiogram pulmonary arterial pressure > 40 mmHg in the entire cohort. Demographic, clinical, and treatment variables were recorded for all patients. Statistical significance was denoted by p values < 0.05. Fifty-nine patients were treated with bosentan a median of 34 months. 13.8% of treated patients had pulmonary hypertension vs 23.7% of untreated patients (p 0.13) during the follow up. In multivariate analysis patients not treated with bosentan had 3.9fold-increased risk of pulmonary hypertension compared with patients under bosentan treatment (p < 0.02). Moreover the percentage carbon monoxide diffusing capacity (DLCO) in bosentan treated patients did not decrease from baseline to the end of follow-up (61.8±14% vs 57±20.1%, p = 0.89). We concluded that Systemic sclerosis patients with digital ulcers treated with bosentan seems to have less risk to develop pulmonary hypertension and to stabilize DLCO

## Introduction

Systemic Sclerosis (SSc), also known as scleroderma, is an autoimmune disease characterized by microangiopathy, fibrotic disfunction and immune disregulation. Currently we have strong evidence of microvascular involvement at the beginning of the disease [1]. The most important

**Funding:** The authors received no specific funding for this work.

**Competing interests:** The authors have declared that no competing interests exist.

vascular complications are digital ulcers (DUs) and pulmonary hypertension (PH). DUs are a frequent severe vascular complication with a prevalence of 40–50% [2–5] being an important cause of disability [6–8]. The endotelin receptor antagonist bosentan (BOS) demonstrated to prevent DUs episodes in two randomized controlled trials (RCT) [6,9].

Pulmonary Arterial Hypertension (PAH) is the most frequent cause of PH in SSc and a leading cause of death [10,11]. A recent meta-analysis showed a 9% prevalence of PAH in SSc [12]. The prognosis of PAH improved since the emergence of new treatments with one, two and three years survival rates of 78%, 58% and 47% respectively [13]. Bosentan has been demonstrated to improve the New York Heart Association Functional Class (NYHA-FC), exercise capacity, haemodynamic parameters, survival [14] and to delay the clinical worsening of patients with PAH [15–17]. Bosentan was also demonstrated to be useful in the initial stages of PAH [17] and it has been indicated to treat PAH in NYHA-FCII/III. It is noteworthy that compared with idiopathic PAH the survival in SSc is worse [18] and that screening programs of PAH for SSc patients were associated with a better outcomes [13].

In scleroderma is hypothesized that therapies used to treat different vascular involvements could be useful to prevent other complications [1]. In SSc exist a shared damage mechanisms [19]. Some studies investigate the clinical vascular complications in patients with and without DUs. The Canadian Scleroderma Research Group investigated in 998 patients the findings associated with DUs [20]. They found no association between DUs and PAH. No association was found between DUs and PAH in a Spanish study with 1326 SSc patients (552 with previous history of DUs) [21]. On the other hand, the German Network in SSc found PAH as a risk factor to present DUs [22]. The ItinérAIR-Sclérodermie French cohort (599 SSc patients) found not increased frequency of PAH in patients with history of DUs, but digital ulceration was associated in patients with diffusing capacity for carbon monoxide (DLCO) of <60% [23]. This was an important finding because a reduced DLCO without impairment in pulmonary function may be a surrogate marker of vasculopathy.

Our purpose was to determine if bosentan could prevent PH in a large cohort of patients with DUs.

## Material and methods

A retrospective case-control study that was carried out in 237 SSc patients with previous history of DUs from cohorts of two scleroderma reference centers. All patients full-filed ACR/EULAR 2013 classification criteria for scleroderma [24] and were followed since January 1980 to December 2012 or up the lack of the follow-up or patient's decease. Case definition was for all patients who received bosentan treatment to prevent DUs for at least one month and controls were those naïve for bosentan.

All patients had documented at least one DU. We excluded patients without history of DU or with PH/PAH diagnosed previously.

### Definition of DU

We evaluated patients rather with the presence of active DU or with a history of DU related to digital ischemia, according to current definition of DU [25]. Ulcers overlying bony prominence secondary to trauma at the site of joint contractions, areas of calcium extrusion and skin fissures were excluded.

### Definition of pulmonary hypertension

Our study was performed in patients from two historical cohorts, we did have not right heart catheterization (RHC) of most patients available, thus we assumed an echocardiographic

definition of PH. We considered that patients with estimated systolic pulmonary arterial pressure (sPAP) > 40mmHg are at higher risk to present PH.

### Dataset

We obtained epidemiological and clinical parameters of the patients in the study. In those exposed to bosentan, we analysed the use, dose and treatment duration. Echocardiography and pulmonary function test of all patients were both recorded from the beginning till the end of the study. Data of the patients were anonymized previously to start the study.

### Statistical analysis

Descriptive statistics of all available data were made. We detailed the number of cases and percentage in qualitative analysis and average, median, standard deviation, 25–75 percentile and 95% confidence interval (CI) for quantitative data.

Fisher's exact test to analyse qualitative data and Student test to analyse quantitative independent variables were used.

To identify the sensitivity change of echocardiographic and pulmonary function test from the beginning to the end of follow-up for all patients we applied T test for paired data and McNemar test. To compare the change between case and control groups the U Mann-Whitney and exact Fisher test were applied.

A multivariate logistic regression model (stepwise regression) were used between the presence of pulmonary hypertension (dependent variable) respect to the administration of Bosentan and those variables with bivariate association ($p < 0.05$) in patients treated or not with bosentan (independent variables), for detect possible confounding factors between the association of pulmonary hypertension and administration of bosentan. Taking into consideration our design a post-hoc analysis was not performed. Once the variables with a statistically significant association ($p < 0.05$) were obtained with respect to the presence of PH *Odds Ratio* (OR) was calculated for each variable with respect to the risk of presenting pulmonary hypertension according to the definition of the study. A p value under 0.05 was considered significant. All statistical analyses were carried out with the statistics software SAS program V9.2SAS Institute Inc., Cary, NC, USA.

### Ethics statement

Approval of Ethics Committee was not necessary based on the Spanish Biomedical Investigation Law 14/2007 (Spanish Ministry of Health) by the characteristics of the our study because data was retrieved using previous existing information present in the historical database from the two cohorts fully anonymized previously to our analysis. Authors had access to the data from March 2013 to July 2013 included.

## Results

After to exclude fifteen patients with previous suspicion (estimated sPAP > 40 mmHg) or PH/PAH diagnosis, a total of 222 patients were included in the analysis. 201 (91%) were women with a mean age of 63.9 ± 19.6 years and with a median age at first SSc manifestation of 40 (95%CI 30–52) years. 138 patients were classified as limited cutaneous subset (lcSSc). Baseline characteristics are shown in Table 1. The bosentan group included 59 patients (26.6%) and 163 were considered as a control group. An estimated sPAP > 40 mmHg was observed in 21% of patients and RHC was performed in 14 patients. We observed a higher percentage of patients with left ventricle diastolic dysfunction (13.9% vs 32.2%; p<0.001) and higher tricuspid regurgitation velocity (TRV) (1.93±1.05 vs 2.42±0.87 m/s; p<0.05) at the end of the follow-up.

Values of forced vital capacity (FVC), carbon monoxide diffusion capacity (DLCO) and DLCO adjusted by alveolar volume (KCO) was also impaired (83.5±19.2% vs 77.8±21.2%; p < 0.0001, DLCO 64.4±18.5% vs 59.2±19.9% KCO 75±20% vs 67.5±18.7; p <0.0001). We did not find differences in FVC/DLCO ratio at end of study.

## Treatment characteristics

59 patients were at least one month under bosentan treatment, with a median of 34 (95% CI 5–59) months. The most conventional dose was 250mg bid (60%). 35.6% of patients had at least one adverse effect (Fig 1) with a withdrawal in 27.1% of patients. The other concomitant vasodilators were calcium channel blockers in 85% of patients, phosphodiesterase type 5 (PDE-5) inhibitors in 14.3% of patients and 12.7% of patients were treated with prostanoids.

## Clinical characteristics

No differences were found between the proportion of patients treated with bosentan in both DUs patients cohorts (35% vs 24%; p = 0.14). The percentage of patients with diffuse cutaneous involvement (dcSSc) was higher in bosentan group (47.5% vs 22.7%; p<0.01). Not significant differences were observed within other organs involvements between treated or no treated.

**Pulmonary hypertension in patients with or without bosentan treatment.**   During the follow-up, 13.8% of treated patients with bosentan and 23.7% of no treated developed pulmonary hypertension (PH) based on echocardiography definition (OR = 0.52; CI95% = 0.22–1.19) (Fig 2). Although this data did not reach the statistical significance (p >0.05).

**Right Heart Catheterism (RHC).**   Only had RHC data of 14 patients in both retrospective cohorts. PAH was confirmed in 7 patients. We did not observe haemodynamic differences between patients with or without treatment.

**Risk to develop PH.**   Interestingly, patients without history of bosentan treatment had 3.9fold-increased risk to present PH compared with patients treated with an *Odds Ratio* (OR) of 3.91 (IC95%:1.3–11.6; p<0.02). The risk of PAH observed in patients who did not take PDE5-inhibitors were similar (OR 3.74; IC95%: 1.2–11.5; p<0.03) and less evident in patients who had never taken prostanoids (OR 2.65; IC95%:1.0–7.0; p<0,05). No differences were found among other treatments (Fig 3).

**Pulmonary function test and echocardiographic findings.**   We found that %DLCO predicted levels was stabilized in the bosentan group. In contrast the control group presented lower levels of %DLCO at the end of the follow-up (65.5±20.2% vs 60.5±19.9%; p<0.04) (Fig 4). In echocardiographic parameters we found a trend with more LV diastolic disfunction in the control group. We did not find any difference in other echocardiogram values (Table 2).

## Discussion

Microvascular involvement is the main feature in SSc, and overexpression of type 1 endotelin (ET-1) is involved as a trigger for complications [26]. It is reasonable to consider that blocking ET-1 could be useful to treat SSc complications [15]. We found a possible protective effect of bosentan to develop PH in SSc. Patients with bosentan presented a prevalence of suspected PH of 13.8% vs 23.7% of patients without after adjusting to confounding factors. No statistical significance was acchieved, but taking into consideration that we used Fisher exact test, a 10% difference between groups could be clinically outstanding. We estimated a number of 240 patients in each group to obtain a 80% of statistical power to find differences between groups. The multivariate results support a protective effect of bosentan. We found that patients treated with bosentan had 3.9 time-fold decrease to develop suspected PH by echo during the follow-up. Moreover we found that treated patients did not present an impairment of DLCO levels.

This result is remarkable because worse values of % DLCO are associated with PAH higher risk [27,28]. To stabilize % DLCO values could be protective against PAH development. Echo parameters did not show significant differences between both groups. Considering all our results together a randomized controlled trial to confirm our data is strictly essential.

The preventive effect of bosentan in SSc vascular complications has been only analyzed in two randomized controlled trials (RCT) involving DU prevention [6,9] that supports the use to prevent DUs in patients with SSc and previous history of DU. None other vascular complications were analyzed in both trials.

None RCTs that analyze the use of drugs for primary prevention of vascular complications in SSc exist. Our present retrospective case-control study is the first to demonstrate the lower risk to develop PAH (estimated by echocardiogram) in patients treated with bosentan. Nevertheless, other authors analyzed if bosentan could be a protective effect of PAH in SSc.

Only one study analyses the effect of bosentan to prevent PAH in the literature. Murdaca *et al*. followed 69 patients with SSc 25 of them with DUs treated with bosentan and compared them with the remaining 44. They did not found cases of PAH in bosentan group and 7 PAH-SSc cases in the control group. Indeed the investigators found an increase in echocardiography parameters in the control group and lower values in bosentan group and better parameters of terminal N of brain natriuretic peptide (NT-proBNP) at the end of the follow-up [29]. Some differences exist between those results and our study: firstly, no details of the number of patients with DUs in control group were given and they did not give a full comparison with the other treatments. Moreover, they analyzed the effect of bosentan in NT-proBNP values but they did not study the effect of the treatment in %DLCO, a powerful predictive factor for PAH.

Romaniello *et al*. studied PAH and sPAP on echocardiogram in 54 SSc with DUs treated with bosentan without previous echocardiografic evidence of PH [30]. They did RHC in patients with sPAP $\geq$ 45mmHg and compared sPAP of patients at the beginning and at the end of the study. They did not find differences in sPAP values and only one patient was diagnosed of PAH by RHC. Similarly to our results, they did not find differences in echocardiogram sPAP, but they did not compare treated group with control group and they did not analyze the effect of concomitant medication. In our study we lowered the bias produced by other treatments by control group and multivariate analysis.

An observational retrospective study with 89 patients with SSc in treatment with bosentan to prevent DUs was published recently [31]. The authors found a lower prevalence of PH (3,4%) compared with other cohorts. This could imply a preventive effect of bosentan in PH. Unlike our study, the authors did not analyze the PH presence or echo data before treatment initiation and they did not have control group. Other authors compared 30 PAH-SSc patients treated with bosentan at least 6 months versus 30 SSc patients without PAH and without bosentan use [32] and demonstrated a significant lower frequency of DU in bosentan group (20% vs 53.3%; p<0.01).

More interesting has appeared to explore the effect of bosentan treatment to stabilize % DLCO. A recent work in 10 PAH-SSc patients treated with bosentan authors found no decrease of %DLCO during treatment period [33]. Those data are similar with our findings. Seibold *et al*. found no differences in %DLCO between bosentan vs placebo group [34], but the design was made to study patients with interstitial lung disease that had low initial levels of %DLCO (%DLCO bosentan: 45.3±12.5 vs %DLCO placebo: 45.1±12.4), so this finding complicate the interpretation.

The protective effect in SSc-PAH development was studied with other drugs. Caramaschi *et al*. analyzed retrospectively the incidence of severe vascular complications in 115 patients (10 of whom with bosentan) treated with iloprost for Raynaud's phenomenon or DUs at least

of 3 years [35]. They found an incidence rate of PAH during the follow-up lower than incidence published previously [36]. Thus, a possible protector effect of iloprost in development of PAH in SSc could exist. In our study, we found an OR 2.7 times fold-risk of PH in patients without prostanoids treatment. It is remarkable that in Caramaschi *et al.* study did not have a control group, and they had a group of patients treated with bosentan also who were not analyzed. In addition, they did not study DLCO. Our data support a possible protector effect of prostanoids by the multivariate analysis. Nevertheless, not all data published are favorable for a protective effect of prostanoids. Airò *et al.* did not find differences in PH existence in 56 patients treated with iloprost when compared with 56 matched controls [37].

Another secondary endpoint in our study was if PDE-5 inhibitor treatment could be protective against PAH development. In our study the OR to develop suspected PH was 3.75 times lower in comparison with no history of taking sildenafil. We had only 8 patients in the sildenafil group who never took bosentan. Unlike to bosentan there are no studies in the literature that demonstrated a protective effect of sildenafil in a PAH.

Our study presents several limitations. First the retrospective design limits the collection data, therefore we can not determine the effect of intermediate variables during the follow-up. Another limitation is the scarce epidemiological data and the no standardized length of the follow-up of the patients. Other confounding factor is the PH definition by echo. We know that the gold standard is RHC, but unfortunately we do not have RHC of all patients by our retrospective design. Some parallel studies between RHC and echo demonstrated that sPAP and TRV correlates with haemodynamic parameters [38], but we cannot exclude false positive cases. We only include patients with DUs for two interesting reasons: the indication of use bosentan in Spain (prevention of DU or treatment of PAH) and the intention to have homogeneous groups. Another lack for our study is that we did not analyze serological biomarkers than could be useful in PAH [39–41]. Sadly, we did not have values of most patients in both historical cohorts. It has been demonstrated a inverse correlation between DLCO and NT-proBNP [42–44] and taking into consideration the current evidence about DLCO role in PAH prognosis [27,41,43] our data suggest that DLCO stabilization in the case group could be also any kind of effect in NT-proBNP values. As a major concern, we fully consider the importance of DLCO as a predictor factor for DU and PAH in SSc. Steen *et al.* matched 106 patients who had PAH-SSc with 106 SSc patients [27]. They found that DLCO levels can predict the development of PAH-SSc. It suggests that a stabilization of DLCO could influence to develop PAH.

Finally, another limitation is that our group did not study the survival fact that it would be interesting.

In summary, our study demonstrates that patients who initiated bosentan to prevent DU have a lower risk to present PH estimated by echocardiography and stabilizes DLCO. We observed a modest effect with prostanoids and sildenafil as well. A randomized control trial is warranted to demonstrate a protective effect of these specific drugs.

## Supporting information

**S1 File.**
(XLSX)

## Author Contributions

**Conceptualization:** Ivan Castellví, Carmen Pilar Simeón, Jordi Casademont, Hèctor Corominas.

**Data curation:** Ivan Castellví, Carmen Pilar Simeón, Monica Sarmiento.

**Formal analysis:** Ivan Castellví, Carmen Pilar Simeón, Monica Sarmiento, Hèctor Corominas.

**Investigation:** Ivan Castellví, Carmen Pilar Simeón, Monica Sarmiento, Jordi Casademont.

**Methodology:** Ivan Castellví, Carmen Pilar Simeón, Jordi Casademont, Hèctor Corominas.

**Project administration:** Ivan Castellví, Carmen Pilar Simeón.

**Resources:** Ivan Castellví, Carmen Pilar Simeón.

**Software:** Ivan Castellví.

**Supervision:** Ivan Castellví, Carmen Pilar Simeón.

**Validation:** Ivan Castellví, Carmen Pilar Simeón.

**Visualization:** Ivan Castellví, Carmen Pilar Simeón.

**Writing – original draft:** Ivan Castellví.

**Writing – review & editing:** Ivan Castellví, Carmen Pilar Simeón, Monica Sarmiento, Jordi Casademont, Hèctor Corominas, Vicenç Fonollosa.

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
