## [Decision Letter · Decision Letter 0]

2 Sep 2020

PONE-D-20-18616

Protective effect of bosentan against pulmonary hypertension development in systemic sclerosis patients with digital ulcers

PLOS ONE

Dear Dr. Castellví,

Thank you for submitting your manuscript to PLOS ONE. After careful consideration, we feel that it has merit but does not fully meet PLOS ONE’s publication criteria as it currently stands. Therefore, we invite you to submit a revised version of the manuscript that addresses the points raised during the review process.

We look forward to receiving your revised manuscript.

Kind regards,

Luca Navarini

Academic Editor

PLOS ONE

Journal Requirements:

2. Please refer to any post-hoc corrections to correct for multiple comparisons during your statistical analyses. If these were not performed please justify the reasons. Please refer to our statistical reporting guidelines for assistance (https://journals.plos.org/plosone/s/submission-guidelines.#loc-statistical-reporting).

3. Please also clarify whether the data you used for this study was de-identified or anonymized before you had access to it.

6. Please amend your list of authors on the manuscript to ensure that each author is linked to an affiliation. Authors’ affiliations should reflect the institution where the work was done (if authors moved subsequently, you can also list the new affiliation stating “current affiliation:….” as necessary).

Reviewers' comments:

Reviewer's Responses to Questions

**Comments to the Author**

1. Is the manuscript technically sound, and do the data support the conclusions?

Reviewer #1: Partly

Reviewer #2: Partly

2. Has the statistical analysis been performed appropriately and rigorously? 

Reviewer #1: Yes

Reviewer #2: N/A

3. Have the authors made all data underlying the findings in their manuscript fully available?

Reviewer #1: Yes

Reviewer #2: Yes

4. Is the manuscript presented in an intelligible fashion and written in standard English?

Reviewer #1: Yes

Reviewer #2: Yes

5. Review Comments to the Author

Reviewer #1: The authors report an important research regarding “Protective effect of bosentan against pulmonary hypertension development in systemic sclerosis patients with digital ulcers.”

The major weaknessess are lack of novelty and right heart catheterization to better define pulmonary artery hypertension.

Minor points:

- The authors report in the introduction “…Pulmonary Arterial Hypertension (PAH) is the most frequent cause of PH in SSc and a leading cause of death” buti t is not clear in the article if they are evaluating PAH or PH. Moreover any patients showed left ventricular dysfunction, are those patients affected by PH group II? What about group III?

-There were patients who were treated only with f PDE-5 inhibitor for treatment of digital ulcers?

Reviewer #2: A retrospective study investigating the role of bosentan in the prevention of Systemic Sclerosis- Pulmonary Arterial Hypertension (SSc-PAH) is of high interest even though there are lots of bias explaining that the title should be less affirmative. There are several issues/comments:

Major comments:

1.Title:

This study did not demostrates the effect of bosentan in the prevention of SSc-PAH. It described the occurrence of PAH in different groups of patients taken bosentan or not. The observational nature of the study does not allow to prospect any cause/effect relationship. Well-designed randomised controlled trials (RCTs) are needed to either support or refuse this hypothesis.

2.Material and methods (page 5):

-The variable, non-standardised length of follow-up represents a limitation.

-The incidental finding of high systolic pulmonary artery pressure on echocardiography is common and is not enough to diagnose pulmonary hypertension. Conditions associated with high systolic PA pressure on echocardiogram are not necessarily associated with PAH. Have you excluded patients with hemodynamically significant valvular disease, a history of uncontrolled systemic hypertension, hyperlipidaemia, cardiac failure, hepatic failure, diabetes, cerebrovascular diseases, peripheral vascular diseases, coagulopathy, smokers and pregnant or breastfeeding women?

3.Statistical analysis (page 6):

-Please, clarify more clearly which variables are included in multivariate regression analysis.

4.Results (page 7):

-Please specify in the results section why 15 of 237 enrolled patients were excluded from the study.

5.Table 1 (page 9):

- Please indicate the age, the prevalence of anticentromere and anti-topoisomerase I antibodies in both treatment and control group and the respective p-values.

- Although it is known, please explain the acronym dcSSc in the table.

- Please, add the p-value (i.e. 1.00) to the digital ulcers row and edit the term “Digital ulcers” to “Active digital ulcers or history of digital ulcers” as defined in Material and Methods section.

- I suggest using the term Renal involvement instead of SRC.

- Risk factors for SSc-PAH such as ACA, older age, longer disease duration, limited cutaneous SSc subset and presence of ILD might be considered (Y. Jianga et al. Autoimmunity Rev. 2020). I suggest therefore to add disease duration to baseline characteristics.

- About half of the cohort (49.1%) presented ILD. I suggest to specify other treatments (i.e. corticosteroids, immunosuppressive agents, cytotoxic drugs, or antifibrotic drugs) that could affects the study results.

Minor comments:

1.Abstract (page 2 line 3):

- In the Introduction I suggest to add the acronym PH (third line) if you want to use it later in the objective section.

2.Figure 3:

-Please edit the IC of Prostanoids to (1,0-7,0) as reported in the text (page 11).

3.Discussion (page 13):

-Please edit Discusion in Discussion.

6. PLOS authors have the option to publish the peer review history of their article (what does this mean?). If published, this will include your full peer review and any attached files.

Reviewer #1: No

Reviewer #2: No

---

## [Author Response · Author response to Decision Letter 0]

20 Oct 2020

Response to Editor and Reviewers

Dear Dr. Navarini, I hope you are well. First of all I would like to thank you and the reviewers your comments about our manuscript. We have read all the considerations with high interest. We are happy to notify that we modified the manuscript and answered your suggestions in the next lines. Please don't hesitate to contact with us if you need more requirements. 

All the best

Ivan Castellví, MD; PhD.

Response to Editor: 

Journal Requirements:

1. We reviewed all the manuscript to meet PLOS ONE’s style requirements.

2. Please refer to any post-hoc corrections to correct for multiple comparisons during your statistical analyses. If these were not performed please justify the reasons. Please refer to our statistical reporting guidelines for assistance (https://journals.plos.org/plosone/s/submission-guidelines.#loc-statistical-reporting).

2. Done. Taking into consideration our design A post-hoc analysis was not performed. We included those explanation in material and methods

3. Please also clarify whether the data you used for this study was de-identified or anonymized before you had access to it.

3. We clarified it in Material and Methods part of the manuscript. Data of the patients were encrypted before to start our retrospective study. 

 4. Thank you for your reminder. We will upload our minimal data set as a Supporting Information file. 

5. Thank you for your clarification. We modified it. 

6. Please amend your list of authors on the manuscript to ensure that each author is linked to an affiliation. Authors’ affiliations should reflect the institution where the work was done (if authors moved subsequently, you can also list the new affiliation stating “current affiliation:….” as necessary).

6. Corrected. There was a mistake in one of author affiliation (sorry). 

Thank you again.

Response to Reviewers: 

Reviewer #1: 

The authors report an important research regarding “Protective effect of bosentan against pulmonary hypertension development in systemic sclerosis patients with digital ulcers.”

The major weaknessess are lack of novelty and right heart catheterization to better define pulmonary artery hypertension.

Thank you for your comments. It's true that the main weakness of our study was the absence of the majority of RHC to design better than using an estimated sPAP by echocardiography. But our retrospective design and to use a historical cohort (with patients followed previous to current PAH detection recommendations by RHC) limits our work in this point.

Minor points:

- The authors report in the introduction “…Pulmonary Arterial Hypertension (PAH) is the most frequent cause of PH in SSc and a leading cause of death” but it is not clear in the article if they are evaluating PAH or PH. Moreover any patients showed left ventricular dysfunction, are those patients affected by PH group II? What about group III?

- Your comment is true definitely. Some patients can present group II or III (or combined I-II / I-III) of PH in SSc. But it is know that PAH (group I) is the most important cause of PH in SSc and is critical to find or to treat in the first phases of the (PH) disease. On the other hand bosentan only demonstrated be useful to treat group I PH (PAH), and our intention was in this retrospective study to investigate if (as in DUs) it drug can prevent PAH also. But (and in association with your comment previous your minor points) we can not are sure what kind of PH is in our patients without an haemodynamic study. Taking in consideration that we used echocardiography we use PH term. But our intention was evaluate the risk of PAH. Linked with this explanation we investigated and showed in the manuscript the effect of bosentan in other risk factors associated with PAH in SSc (DLCO).

-There were patients who were treated only with f PDE-5 inhibitor for treatment of digital ulcers?

- Thank you for your question. A total of 31 patients were in treatment for DUs with PDE-5 inhibitors. (23 concomitantly in bosentan group and eight in non bosentan group) as we shown in table 1. 

Thank you again for your constructive comments. 

Reviewer #2: 

A retrospective study investigating the role of bosentan in the prevention of Systemic Sclerosis- Pulmonary Arterial Hypertension (SSc-PAH) is of high interest even though there are lots of bias explaining that the title should be less affirmative. There are several issues/comments:

Major comments:

1.Title:

This study did not demostrates the effect of bosentan in the prevention of SSc-PAH. It described the occurrence of PAH in different groups of patients taken bosentan or not. The observational nature of the study does not allow to prospect any cause/effect relationship. Well-designed randomised controlled trials (RCTs) are needed to either support or refuse this hypothesis.

1.Thank you for your observation. we only can agree with you. We changed the title for other more appropriate with our study. 

2.Material and methods (page 5):

-The variable, non-standardised length of follow-up represents a limitation.

- It’s true. We included in the limitations of our study (page 15). 

-The incidental finding of high systolic pulmonary artery pressure on echocardiography is common and is not enough to diagnose pulmonary hypertension. Conditions associated with high systolic PA pressure on echocardiogram are not necessarily associated with PAH. Have you excluded patients with hemodynamically significant valvular disease, a history of uncontrolled systemic hypertension, hyperlipidaemia, cardiac failure, hepatic failure, diabetes, cerebrovascular diseases, peripheral vascular diseases, coagulopathy, smokers and pregnant or breastfeeding women?

- Again you are absolutely on right. We did not exclude the major part of this risk factors by missing data in reviewed database (with an important number of historical patients). We also cited the lack of some epidemiological data in the limitations of our study (pages 15 & 16). We are sorry but we can’t recover this kind of missing data. On the other hand we think that, taking into consideration all those limitations (and the difficult to obtain data about this complication in patients with a rare disease), are relevant to take into account. 

3.Statistical analysis (page 6):

-Please, clarify more clearly which variables are included in multivariate regression analysis.

- We re-edited the explanation with the intention to do this part more clear (page 6). 

4.Results (page 7):

-Please specify in the results section why 15 of 237 enrolled patients were excluded from the study.

- We included the explanation in Results (page 7). 

5.Table 1 (page 9):

- Please indicate the age, the prevalence of anticentromere and anti-topoisomerase I antibodies in both treatment and control group and the respective p-values.

- Done. 

- Although it is known, please explain the acronym dcSSc in the table.

- Done.

- Please, add the p-value (i.e. 1.00) to the digital ulcers row and edit the term “Digital ulcers” to “Active digital ulcers or history of digital ulcers” as defined in Material and Methods section.

- Done.

- I suggest using the term Renal involvement instead of SRC.

- Thank you for the suggestion. Done. 

- Risk factors for SSc-PAH such as ACA, older age, longer disease duration, limited cutaneous SSc subset and presence of ILD might be considered (Y. Jianga et al. Autoimmunity Rev. 2020). I suggest therefore to add disease duration to baseline characteristics.

- Thank you for your suggestion. We included disease duration in table 1.

- About half of the cohort (49.1%) presented ILD. I suggest to specify other treatments (i.e. corticosteroids, immunosuppressive agents, cytotoxic drugs, or antifibrotic drugs) that could affects the study results.

- We added the immunosuppressive agents in the table 1. 

Minor comments:

1.Abstract (page 2 line 3):

- In the Introduction I suggest to add the acronym PH (third line) if you want to use it later in the objective section.

- Thank you for you comment. We changed “PH” for pulmonary hypertension ( in order to avoid acronyms in the abstract). We also modified the abstract according style requirements of PLOS ONE. 

2.Figure 3:

-Please edit the IC of Prostanoids to (1,0-7,0) as reported in the text (page 11).

- Thank you for your observation. Corrected. 

3.Discussion (page 13):

-Please edit Discusion in Discussion.

- We apologize by mistake. Corrected. 

Thank you again for your constructive comments and suggestions.

---

## [Decision Letter · Decision Letter 1]

25 Nov 2020

Effect of bosentan in pulmonary hypertension development in systemic sclerosis patients with digital ulcers

PONE-D-20-18616R1

Dear Dr. Castellví,

We’re pleased to inform you that your manuscript has been judged scientifically suitable for publication and will be formally accepted for publication once it meets all outstanding technical requirements.

Kind regards,

Luca Navarini

Academic Editor

PLOS ONE

Additional Editor Comments (optional):

Reviewers' comments:

Reviewer's Responses to Questions

**Comments to the Author**

1. If the authors have adequately addressed your comments raised in a previous round of review and you feel that this manuscript is now acceptable for publication, you may indicate that here to bypass the “Comments to the Author” section, enter your conflict of interest statement in the “Confidential to Editor” section, and submit your "Accept" recommendation.

Reviewer #1: All comments have been addressed

Reviewer #2: All comments have been addressed

2. Is the manuscript technically sound, and do the data support the conclusions?

Reviewer #1: Yes

Reviewer #2: Yes

3. Has the statistical analysis been performed appropriately and rigorously? 

Reviewer #1: Yes

Reviewer #2: Yes

4. Have the authors made all data underlying the findings in their manuscript fully available?

Reviewer #1: Yes

Reviewer #2: Yes

5. Is the manuscript presented in an intelligible fashion and written in standard English?

Reviewer #1: Yes

Reviewer #2: Yes

6. Review Comments to the Author

Reviewer #1: The paper is improved and all the suggestions have been raised. The paper is well written and the results are important in the management of systemic sclerosis complications

Reviewer #2: (No Response)

7. PLOS authors have the option to publish the peer review history of their article (what does this mean?). If published, this will include your full peer review and any attached files.

Reviewer #1: **Yes: **antonietta gigante

Reviewer #2: No

---

## [Editor Report · Acceptance letter]

1 Dec 2020

PONE-D-20-18616R1 

Effect of bosentan in pulmonary hypertension development in systemic sclerosis patients with digital ulcers 

Dear Dr. Castellví:

I'm pleased to inform you that your manuscript has been deemed suitable for publication in PLOS ONE. Congratulations! Your manuscript is now with our production department. 

Kind regards, 

on behalf of

Dr. Luca Navarini 

Academic Editor

PLOS ONE